# Point-Line Visual Stereo SLAM Using EDlines and PL-BoW

Hanxiao Rong [1], Yanbin Gao [1], Lianwu Guan [1,*], Alex Ramirez-Serrano [2], Xu Xu [1] and Yunyu Zhu [3]

[1] College of Intelligent Systems Science and Engineering, Harbin Engineering University, Harbin 150001, China; ronghanxiao@hotmail.com (H.R.); gaoyanbin@hrbeu.edu.cn (Y.G.); xuxu66@hrbeu.edu.cn (X.X.)
[2] Department of Mechanical Engineering, University of Calgary, Calgary, AB T2N 1N4, Canada; aramirez@ucalgary.ca
[3] First Institute of Oceanography, MNR, Qingdao 266061, China; zhuyy@fio.org.cn
* Correspondence: guanlianwu@hrbeu.edu.cn

**Abstract:** Visual Simultaneous Localization and Mapping (SLAM) technologies based on point features achieve high positioning accuracy and complete map construction. However, despite their time efficiency and accuracy, such SLAM systems are prone to instability and even failure in poor texture environments. In this paper, line features are integrated with point features to enhance the robustness and reliability of stereo SLAM systems in poor texture environments. Firstly, method Edge Drawing lines (EDlines) is applied to reduce the line feature detection time. Meanwhile, the proposed method improves the reliability of features by eliminating outliers of line features based on the entropy scale and geometric constraints. Furthermore, this paper proposes a novel Bags of Word (BoW) model combining the point and line features to improve the accuracy and robustness of loop detection used in SLAM. The proposed PL-BoW technique achieves this by taking into account the co-occurrence information and spatial proximity of visual words. Experiments using the KITTI and EuRoC datasets demonstrate that the proposed stereo Point and EDlines SLAM (PEL-SLAM) achieves high accuracy consistently, including in challenging environments difficult to sense accurately. The processing time of the proposed method is reduced by 9.9% and 4.5% when compared to the Point and Line SLAM (PL-SLAM) and Point and stereo Point and Line based Visual Odometry (sPLVO) methods, respectively.

**Keywords:** stereo SLAM; line detection; entropy scale; visual odometry; loop closure

## 1. Introduction

Simultaneous Localization and Mapping (SLAM) was initially proposed by Smith in 1987 [1]. Since then, diverse methods systems that can simultaneously estimate the position of the onboard sensors and construct the surrounding environment map via the captured scene information have been extensively developed. This has been especially important in the field of robot navigation using diverse types of camera systems (e.g., monocular, stereo and panoramic) [2–5].

In recent years, most of the research has focused on improving the accuracy of point-feature SLAM, and many encouraging breakthroughs have been made [6,7]. Such research includes the development of the ORB-SLAM2 system based on Parallel Tracking and Mapping (PTAM) ideas that improve the place recognition and the loop closure modules [8]. The invariant of ORB features to different viewpoints and illuminations has been employed to improve tracking, mapping, and loop closure. As a result, ORB-SLAM2 has become one of the most popular systems in rich-textured scenes. Despite the success of the ORB-SLAM2 framework, the expanding applications of mobile robots, among many other applications, such as augmented reality and autonomous driving, impose new challenges, which still remain to be resolved, related to low-textured or structured engineered environments [9,10]. Traditional point-based strategies are unstable and even fail in some scenarios where the point features are uneven or not well distributed. Fortunately, Cadena et al. pointed out

that a way to solve this problem is to find an alternative feature that is abundant in the environment [3]. The low sensitivity of line features to light changes and motion blurring is also a prevalent topic in SLAM research. Lu et al. proposed an RGB-D SLAM system that combines line features and depth information to solve the problem of indoor illumination changes [11]. Scaramuzza et al. focused on extracting the vertical lines in the wider field image information captured by the omnidirectional camera to improve the accuracy and robustness of the mobile robot visual odometer system [12,13]. Pumarola et al. improved the initial accuracy of the system by synchronously calculating point and line features in the monocular SLAM initialization thread [14]. Ma et al. utilized vanishing points (VP) to constrain line features, which greatly reduced the mismatching of line features [15]. Despite the improvements, the above line segment detection and matching methods are time-consuming, making the line-based SLAM methods difficult to be processed in real-time.

To enhance the effectiveness of line detection methods, Gomez-Ojeda et al. combined a Line Segment Detector (LSD) with a Line Band Descriptor (LBD) algorithm to develop an enhanced stereo version of Point and Line SLAM (PL-SLAM). Such a method has shown higher translation accuracy than that of ORB-SLAM2 in rich-feature scenes, which proved that stereo systems are more accurate and resistant to interference than other monocular systems in line detection [16]. Berenguer et al. proposed a method to estimate the relative attitude by using Holistic descriptors of the omnidirectional image. This method solves the problem that the omnidirectional image can not deal with the height change of mobile robots, but it requires further research to apply in estimating movements with six degrees of freedom [17]. Inspired by the Scale Invariant Feature Transform (SIFT) algorithm, Li et al. proposed the scale-invariant mean-standard deviation LSD (SMLSD) to extract line features faster without sacrificing detection accuracy [18]. However, Zuo et al. proved that the performance of LSD line detection is still unsatisfactory for real-time applications [19]. There is an urgent need to develop line detection methods that can accurately extract line features at a fast rate regardless of the environment geometrical complexity. To realize the real-time SLAM for point and line features, Zhang et al. proposed the line detector based on Canny edges to obtain line features iteratively [20]. In contrast to [20], Vakhitov et al. focused on improving the accuracy and robustness of line feature extraction by training a deep yet lightweight full-convolutional neural network [21]. Gomez-Ojeda et al. [22] and Luo et al. [23] introduced the Fast Line Detector (FLD) algorithm to reduce the detecting time of the LSD method. The fast detection speed and straightforward logic composition of the FLD approach makes it suitable in the field of point-line SLAM. Although effective, it requires prior information about the scene to determine the needed parameters, which limits its usefulness in prior unknown environments, such as those encountered inside collapsed buildings.

Another hindrance that prevents the line feature from being extensively used is the complex outlier culling technique that makes their implementation nontrivial. Line features have a number of characteristics that change with the angle of view and occlusions, so the method of removing inappropriate features from the detected lines has been a focus of constant research. Shao et al. utilized a coplanar constraint of line features to filter mismatched line features, but the method lacks processing real-time data [24]. Ma et al. trained a general classifier to judge the correctness of any hypothetical matching [25]. This method converts the mismatch elimination into a two-class classification problem, but the performance of the algorithm in unknown scenes still requires further experimental verification. Lim et al. proposed a structural model based on epipolar and vanishing points to remove degenerate LSD line features intended to improve the reliability of line features [26].

Although the above-mentioned approaches present many practical solutions to issues existing in line feature detection and outliers elimination, they are still unable to effectively address the SLAM problem of point-line loop detection. Pumarola et al. proposed that only point features were detected in the PL-SLAM loop closure to reduce the computational complexity [14]. Gomez-Ojeda et al. built a line Bag of Words (BoW) model that had a

parallel relationship with the point BoW to achieve the independent line loop detection [22]. However, the point-line detection method without considering the geometric proximity between point and line features does not significantly improve the accuracy of loop detection. Ma et al. modeled a BoW containing VPs and their connecting lines to improve the accuracy of the SLAM system in the corridor [15]. For the problem of false loop detection caused by the existence of a similar point map, researchers have pointed out that the accuracy and robustness of loop detection can be improved by building a Bag of Words (BoW) model that considers the spatial proximity and co-occurrence information of visual vocabulary [27,28]. Under this context, an effective Point and EDlines SLAM framework, named PEL-SLAM, is proposed to solve the above-mentioned issues, including the line feature detection, line outliers elimination, and loop detection.

Specifically, a point-line stereo SLAM system based on Edge Drawing lines (EDlines) [29] and the improved BoW model is proposed to realize localization and mapping in human-made environments. The proposed approach takes advantage of line features in low-textured environments and PL-BoW for loop detection. Point and line features of the input stereo image are detected by ORB and EDlines, followed by calculating the uncertainty entropy that describes the accuracy of the detected line features. The PL-BoW is then obtained using a PL pair selection technique following the ORB point and LBD line descriptors. Finally, a loop detection mechanism based on PL-BoW, similarity score, and space consistency detection of the keyframe is employed to determine the best matching keyframe. The three main contributions of this paper are:

(1) A stereo SLAM system based on the integration of point and line features. Such a method employs an EDlines algorithm to improve the speed of line feature detector in the front-end of the system. In addition, the comprehensive representation and transformation of line features are also derived.

(2) A method using entropy scale and geometric constraints is proposed to eliminate the outliers of line features. The strategy of removing the mismatched features in the front-end ensures the reliability of the extracted lines without increasing the additional algorithm complexity. The application of this method improves the accuracy of camera pose estimation and map construction.

(3) A novel Point and Line Bags of Word (PL-BoW) model combining the point and line features is proposed to improve the accuracy and robustness of loop detection. Unlike popular methods of evaluating the BoW of point and line features independently, the proposed PL-BoW model takes into account the constraints of the extracted point and line features. Such a model improves the reliability of the loop detection process under the interference of weak texture and light changes, which typically exist in structured engineered environments.

The remainder part of this paper is as follows. Section 2 provides the geometric expression, detection, and matching methods of line features. The graph optimization for line features and the improved loop detection mechanism are presented in Section 3. Section 4 compares and analyzes the experimental results of the proposed algorithm with existing methods. The conclusions of the paper are presented in Section 5.

## 2. Representation and Detection of Line Features

The schematic diagram of the PEL-SLAM system proposed here is illustrated in Figure 1. Since the PEL-SLAM system is improved on the basis of ORB-SLAM2, most modules of the system are the same as the ORB-SLAM2 system except for the green part in Figure 1.

Improvements of the line representation, line feature detection, and line feature matching in the front-end of the proposed method are presented in this section, while improvements of Bundle Adjustment (BA) optimization and loop closure in the back-end of the system are described in the following sections.

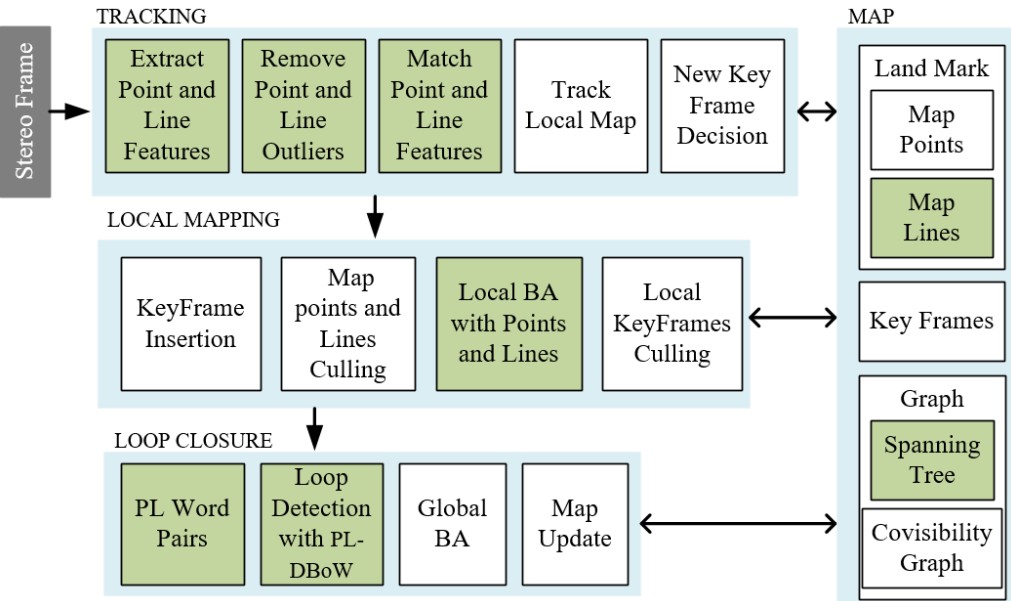

**Figure 1.** The schematic diagram of the proposed PEL-SLAM.

### 2.1. Geometric Representation of Lines

A spatial line can be expressed via its two endpoints in the image plane, $s_k = \begin{bmatrix} u_s & v_s & 1 \end{bmatrix}^T$ and $e_k = \begin{bmatrix} u_e & v_e & 1 \end{bmatrix}^T$ (see Figure 2). This is an essential operation of the visual odometer needed to transform 3D lines in the world reference to the image plane. The Plücker parameter has the characteristic of observability and computational simplicity, and it is used for the transformation and projection of line features. However, the over parameterization of the Plücker parameter hinders the performance of the system in back-end optimization. To address this, an orthonormal representation is introduced for optimization. The spatial line $\mathcal{L}$ in the Plücker coordinate is expressed by $\mathcal{L} = \left( n^T, d^T \right)^T \in \mathbb{R}^6$, where $n \in \mathbb{R}^3$ is the normal vector, and $d \in \mathbb{R}^3$ represents the line direction vector of the plane $\pi_k$. The plane $\pi_k$ is composed of the line $\mathcal{L}$ and the $k_{th}$ camera coordinate origin $C_k$.

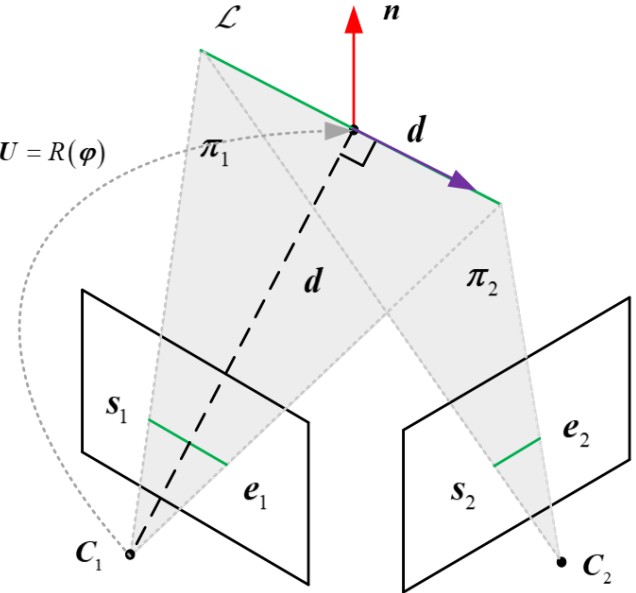

**Figure 2.** Plücker parameters and triangulation of a spatial line.

As shown in Figure 2, the Plücker line coordinate is usually constructed by triangulation of two different camera frames. The parameters of the $\boldsymbol{\pi}_1 = \begin{pmatrix} \pi_x & \pi_y & \pi_z & \pi_w \end{pmatrix}^{\mathrm{T}}$ plane can be determined by $\boldsymbol{s}_1$, $\boldsymbol{e}_1$, and the camera origin $\boldsymbol{C}_1 = \begin{bmatrix} x_1 & y_1 & z_1 \end{bmatrix}^{\mathrm{T}}$ in the world reference as follows:

$$\pi_w = \begin{bmatrix} \pi_x & \pi_y & \pi_z \end{bmatrix} \begin{bmatrix} x \\ y \\ z \end{bmatrix} \tag{1}$$

$$\begin{bmatrix} \pi_x \\ \pi_y \\ \pi_z \end{bmatrix} = [\boldsymbol{s}_1]_\times \boldsymbol{e}_1, \pi_w = \pi_x x_1 + \pi_y y_1 + \pi_z z_1 \tag{2}$$

where the $[]_x$ is the skew-symmetric matrix of a vector in $\mathbb{R}^3$. Other planes (e.g., $\boldsymbol{\pi}_k$) can be obtained following the same calculation.

Given planes $\boldsymbol{\pi}_1$ and $\boldsymbol{\pi}_2$, the coordinate of the line feature in the world reference is determined via the following dual Plücker matrix $\boldsymbol{L}^*$:

$$\boldsymbol{L}^* = \begin{bmatrix} [\boldsymbol{d}]_\times & \boldsymbol{n} \\ -\boldsymbol{n}^{\mathrm{T}} & 0 \end{bmatrix} = \pi_1 \pi_2^{\mathrm{T}} - \pi_2 \pi_1^{\mathrm{T}} \in \mathbb{R}^4 \tag{3}$$

With the Plücker coordinates known, the transformation of the line becomes convenient in 3D Euclidean space. In what follows, we represent a 3D rigid body motion $\boldsymbol{T} = \begin{bmatrix} \boldsymbol{R} & \boldsymbol{t} \\ 0 & 1 \end{bmatrix} \in SE(3)$ by a rotation matrix $\boldsymbol{R} \in SO(3)$ and a translation vector $\boldsymbol{t} \in \mathbb{R}^3$. Given the transformation matrix $\boldsymbol{T}_{cw}$ from the world reference to the camera reference, the corresponding Plücker transformation matrix is defined by:

$$\mathcal{T}_{cw} = \begin{bmatrix} \boldsymbol{R}_{cw} & [\boldsymbol{t}_{cw}]_\times \boldsymbol{R}_{cw} \\ \boldsymbol{0} & \boldsymbol{R}_{cw} \end{bmatrix} \tag{4}$$

Then the Plücker representation of the given line feature can be transformed by:

$$\mathcal{L}^c = \begin{bmatrix} \boldsymbol{n}^c \\ \boldsymbol{d}^c \end{bmatrix} = \mathcal{T}_{cw} \mathcal{L}^w \tag{5}$$

After calculating the representation of the line feature in the camera reference, the spatial line can be projected to the image plane through:

$$\boldsymbol{I} = \begin{bmatrix} l_1 \\ l_2 \\ l_3 \end{bmatrix} = \mathcal{K} \boldsymbol{n}^c = \begin{bmatrix} f_y & 0 & 0 \\ 0 & f_x & 0 \\ -f_y c_x & -f_x c_y & f_x f_y \end{bmatrix} \boldsymbol{n}^c \tag{6}$$

where $\mathcal{K}$ denotes the projection matrix of a given line, and $f_x, f_y, c_x, c_y$ represent the intrinsic parameters of the calibrated camera. It should be noted that when a line feature is projected onto a normalized image plane, $\mathcal{K}$ is an identity matrix.

Since a spatial line in 3D space only has four degrees of freedom, using the Plücker representation increases the computational complexity of the optimization process. The de-coupled orthonormal representation $(\boldsymbol{U}, \boldsymbol{W}) \in SO(3) \times SO(2)$ employed in the unconstrained optimization problem is obtained from the *QR* decomposition of the Plücker coordinates.

$$\begin{bmatrix} \boldsymbol{n} & \boldsymbol{d} \end{bmatrix} = \begin{bmatrix} \frac{\boldsymbol{n}}{\|\boldsymbol{n}\|} & \frac{\boldsymbol{d}}{\|\boldsymbol{d}\|} & \frac{\boldsymbol{n} \times \boldsymbol{d}}{\|\boldsymbol{n} \times \boldsymbol{d}\|} \end{bmatrix} \begin{bmatrix} \|\boldsymbol{n}\| & 0 \\ 0 & \|\boldsymbol{d}\| \\ 0 & 0 \end{bmatrix} = \boldsymbol{U} \begin{bmatrix} w_1 & 0 \\ 0 & w_2 \\ 0 & 0 \end{bmatrix} \tag{7}$$

Furthermore, $U$ and $W$ are expressed by:

$$U = R(\boldsymbol{\varphi}) = \left[ \begin{array}{ccc} \dfrac{n}{\|n\|} & \dfrac{d}{\|d\|} & \dfrac{n \times d}{\|n \times d\|} \end{array} \right] \tag{8}$$

$$W = W(\phi) = \frac{1}{\sqrt{\|n\|^2 + \|d\|^2}} \left( \begin{array}{cc} \|n\| & -\|d\| \\ \|d\| & \|n\| \end{array} \right) \tag{9}$$

where $\boldsymbol{\varphi} = \left[ \begin{array}{ccc} \varphi_1 & \varphi_2 & \varphi_3 \end{array} \right]^{\mathrm{T}}$ denotes the angle of rotation from the camera reference to the line reference (see Figure 2). Since $W$ implies one-dimensional scale information, the minimum parameterization of a spatial line can be defined by a four-dimensional vector $\mathcal{O} = \left[ \begin{array}{cc} \boldsymbol{\varphi}^T & \phi \end{array} \right]^T \in \mathbb{R}^4$. Once the orthogonal representation of the optimized feature is obtained, the corresponding Plücker coordinates are calculated by:

$$\mathcal{L}' = \left[ \begin{array}{cc} w_1 \boldsymbol{u}_1^T & w_2 \boldsymbol{u}_2^T \end{array} \right]^T = \frac{1}{\sqrt{\|n\|^2 + \|d\|^2}} \mathcal{L} \tag{10}$$

in which $\boldsymbol{u}_i$ denotes the $i_{th}$ column of $U$.

### 2.2. Extraction and Description of Line Features

The line segment detector LSD, which gives an accurate line detection and requires no parameter tuning (widely applied in state-of-the-art PL-SLAM systems), is designed to extract line segments on noisy images with sub-pixel detection accuracy [30]. However, the computational complexity of LSD causes extra time spending on the system in the front-end feature extraction. Since the limitation of LSD directly affects the tracking and matching of line features, this aspect leads to the failure of the visual odometer.

To improve the real-time performance of the line feature extraction, the EDlines algorithm is used instead of the LSD to detect line features from images captured by the stereo camera. The EDlines algorithm has been proven to run faster than the LSD, while its output tends to contain irrelevant lines [15]. The experiment in [31] shows that EDlines can achieve a similar performance when compared with LSD without sacrificing the detection accuracy by using an appropriate line outlier elimination method.

In this work, the individual performance of the LSD and EDlines mechanisms was validated on three different scenes. Referring to Figure 3, the median three figures show the line features detected by LSD, whereas the three figures on the right side of the figure show the detected results of EDlines. The results show that both detecting algorithms are able to detect the line features of interest that correspond to real straight lines in the given environments. However, it is noted that EDlines is more likely to detect curves that are near straight lines. The detected curves may degrade the performance of the system. However, these curves usually possess bad spatial positions and thus can be easily recognized from different camera views. Thus, from the testing performed, it was identified that it is possible to use EDlines to achieve effective line detection without corrupting the systems by leveraging some line renting strategies.

To make the line features detected by EDlines more recognizable, the LBD descriptors [32] are computed to represent each line feature. Similar to the ORB descriptor for point feature, the LBD descriptor contains geometric attributes and appearance descriptions of the corresponding line features. For two consecutive stereo frames, the similarity of line features is measured by calculating the consistency of LBD descriptors between line pairs.

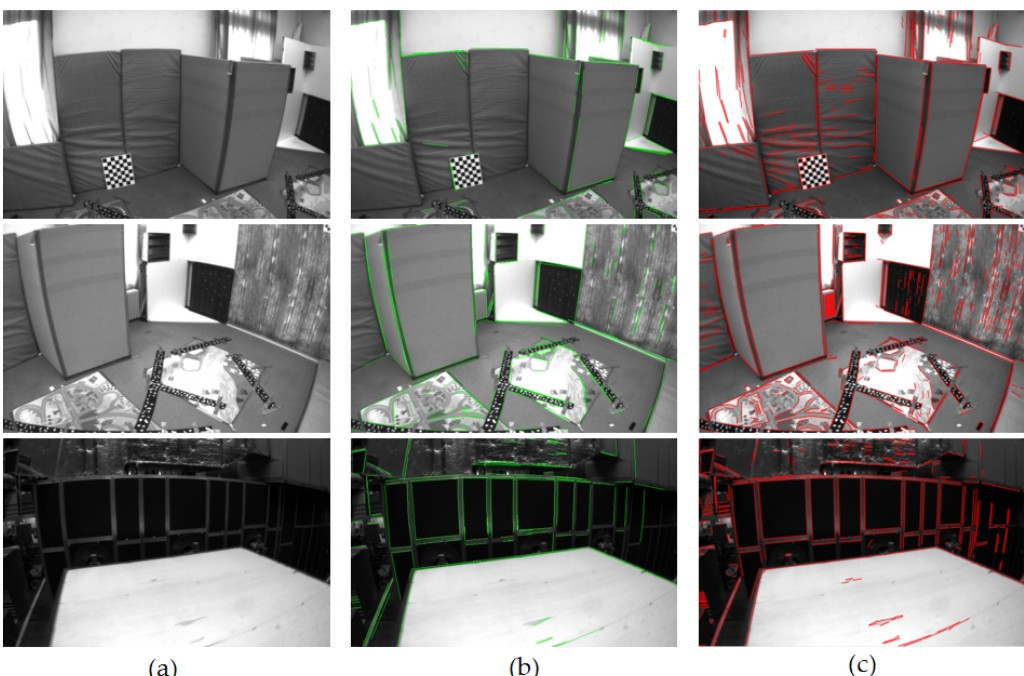

(a) (b) (c)

**Figure 3.** The extracted lines of LSD and EDlines in the EuRoC dataset [33]. (**a**)The original images. (**b**) The green lines in the left column are the line features detected by LSD. (**c**) The red lines in the right column are the line features detected by EDlines.

### 2.3. Extraction and Description of Line Features

After exacting line features, LBD descriptors are generated to describe the corresponding line features in two consecutive captured adjacent frames. The similarity of descriptors is calculated to find the matching line features. The extracted line feature is considered to be a good match only if such feature is the best match in both images of a stereo frame. Before eliminating outliers, the preprocessing of matched pairs is required to improve the accuracy of the system matching. Given the occlusion and perspective changes that might exist in real-world environments, in this work, the line pair is not considered a match if their lengths are more than twice as different. At the same time, the line pair is considered to be mismatched if the distance between the midpoints of the two lines on the image is greater than a given threshold.

For a stereo frame, a spatial line is represented as $l_l$ and $l_r$ in the left and right images, respectively, where $l_r$ can be represented as $\begin{pmatrix} l_{r1} & l_{r2} & l_{r3} \end{pmatrix}^T$ by the homogeneous coordinates of its endpoints. Given a stereo frame captured by a stereo camera, the matching points in the two rectified images have the same horizontal position. Therefore, the endpoints in the right image can be obtained through $l_r$ and line $l_{v=v'}$, as illustrated in Figure 4.

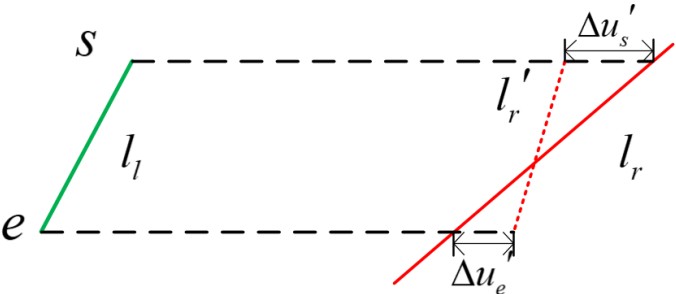

**Figure 4.** The line model in a stereo frame.

The endpoints of the line $l_r$ can be denoted by the endpoints $s = \begin{pmatrix} u_s & v_s \end{pmatrix}^{\mathrm{T}}$ and $e = \begin{pmatrix} u_e & v_e \end{pmatrix}^{\mathrm{T}}$ of the left line.

$$u'_s = -\frac{l_{r2}v_s + l_{r3}}{l_{r1}}, u'_e = -\frac{l_{r2}v_e + l_{r3}}{l_{r1}} \tag{11}$$

The depth parameters of $s$ and $e$ are calculated by the disparities as follows:

$$d_s = \frac{bf_x}{\Delta u'_s}, d_e = \frac{bf_x}{\Delta u'_e} \tag{12}$$

where $\Delta u'_s = u_s - u'_s$, $\Delta u'_e = u_e - u'_e$, and $b$ denotes the baseline of the stereo camera. Then the 3D position of $P_s$ is obtained by the back-projection of the endpoint $s$ as follows:

$$\boldsymbol{P}_s = \begin{bmatrix} X_s \\ Y_s \\ Z_s \end{bmatrix} = \begin{bmatrix} (u_s - c_x)d_s/f_x \\ (v_s - c_x)d_s/f_y \\ d_s \end{bmatrix} \tag{13}$$

Since line features are more sensitive to image noise and mismatching than point features, the uncertainty of line features is modeled to quantify the reliability of line features. Considering the spatial properties of line features, the covariance propagation method is used to construct the uncertainty matrix of the endpoint, which can be expressed as follows:

$$\boldsymbol{\Sigma}_{P_s} = \boldsymbol{J}_{P_s} \mathrm{cov}(\boldsymbol{s}) \boldsymbol{J}_{P_s}^{\mathrm{T}} \tag{14}$$

in which $\boldsymbol{J}_{P_s}$ represents the Jacobian matrix of $\boldsymbol{P}_s$, and $\mathrm{cov}(s)$ is the uncertainty matrix of $\boldsymbol{s}$. Based on the properties of $\boldsymbol{p}_s$ in the image plane, $\mathrm{cov}(s)$ is modeled as a bi-dimensional Gaussian set with standard deviation $\sigma_u = \sigma_v = 1$ pixel. The matrix $\boldsymbol{J}_{P_s}$ can be derived from Equation (13) as follows:

$$\boldsymbol{J}_{P_s} = \frac{\partial \boldsymbol{P}_s}{\partial \boldsymbol{s}} = \begin{bmatrix} d_s/f_x & 0 \\ 0 & d_s/f_y \\ 0 & 0 \end{bmatrix} \tag{15}$$

Due to the fact that the scale of uncertainty between image pairs varies, the values are not directly comparable. The entropy of multivariate normal distribution is thus introduced to abstract the uncertainty in the covariance matrix into a scalar value, which is defined as:

$$H_s = 0.5m(\ln 2\pi + 1) + 0.5\ln(|\boldsymbol{\Sigma}_{P_s}|) \tag{16}$$

where m is the dimension of $\boldsymbol{P}_s$. The uncertainty entropy of $\boldsymbol{P}_e$ can be calculated in the same way.

In the methods of [20,24], the processing of outliers is to remove matching pairs that do not meet a set constant threshold. However, we found that a constant threshold is not applicable because the uncertainty depends on the changes of motion and scene. A threshold determination method based on the uncertainty entropy of all line features of the current frame is thus applied to the outliers eliminating process. After calculating the average entropy $\bar{H}_s$ for all line features of the current frame, the threshold $Td_c$ is set to $0.85 \bar{H}_s$ according to experiments. If the uncertainty entropy $H_s$ and $H_e$ of a spatial line reconstructed from the stereo frame are greater than $Td_c$, the line can be regarded as an accurate line feature, and vice versa.

## 3. Bundle Adjustment and Loop Closure with Points and Lines

The BA module in the back-end optimization consists of two main aspects: local BA of local mapping thread and global BA of loop closure thread. The original BA module of ORB-SLAM2 includes variables of camera poses and point landmarks. However, the addition of line features complicates the optimization process as a minimization process

of the cost function in the co-visibility graph needs to be performed. Therefore, a graph BA optimization strategy considering line and point features is adopted in this paper. In addition, the accuracy of the loop closure depends on the calculation of the global BA and the loop detection. A novel point and line BoW is thus proposed to improve the stability and accuracy of the loop detection.

*3.1. Graph Optimization with Point and Line Features*

Due to the fact that the length and angle of the same line in the two images are different, projection errors cannot be obtained directly from the corresponding two frames. In this work, the projection errors are computed by re-projecting the matched lines from the world reference back to the current image reference. Given the spatial line $\mathcal{L}$ in the world reference and its orthogonal expression $\mathcal{O}$, the corresponding line feature is transformed to the camera frame of reference by $T_{cw}$. Then the projected line $\boldsymbol{I}^{c_i}$ is obtained by projecting the line features onto the normalized plane of the current frame. The reprojection error is then defined as the distance between the projected line $\boldsymbol{I}^{c_i}$ and the endpoints of the detected line $\boldsymbol{l}^{c_i}$, which is computed via Equation (17).

$$\boldsymbol{r}_l = \left[ \begin{array}{c} d\left(\boldsymbol{l}_s^{c_i}, \boldsymbol{I}^{c_i}\right) \\ d\left(\boldsymbol{l}_e^{c_i}, \boldsymbol{I}^{c_i}\right) \end{array} \right], d(\boldsymbol{s}, \boldsymbol{I}) = \frac{\boldsymbol{s}^{\mathrm{T}}\boldsymbol{I}}{\sqrt{l_1^2 + l_2^2}} \tag{17}$$

where $\boldsymbol{l}_s^{c_i}$ and $\boldsymbol{l}_e^{c_i}$ are the endpoints of the detected line $\boldsymbol{l}^{c_i}$. With the reprojection error $\boldsymbol{r}_l$ defined, a global loss function $C$ containing point and line features is formulated as:

$$C = F_{ep} + F_{el} = \sum_{i,k} \rho_l \left( \boldsymbol{r}_l^{\mathrm{T}}(i, k) \Sigma_l^{-1} \boldsymbol{r}_l(i, k) \right) + \sum_{i,j} \rho_p \left( \boldsymbol{r}_p^{\mathrm{T}}(i, j) \Sigma_p^{-1} \boldsymbol{r}_p(i, j) \right) \tag{18}$$

where $\rho$ denotes the robust Huber cost functions, and $\Sigma_l^{-1}$, $\Sigma_p^{-1}$ denote the inverse information matrices of the reprojection error of points and lines, respectively. Compared to the original loss function of the ORB-SLAM2 system, the reprojection error function $F_{el}$ about the line feature is added to Equation (18). The camera pose and landmark are calculated by minimizing the loss function $C$. The Jacobians of $\boldsymbol{r}_p$ are already derived in ORB-SLAM2, and the Jacobians $J_l$ concerning camera pose and line landmark can be expressed by the following matrix chain multiplication:

$$J_l = \frac{\partial \boldsymbol{r}_l}{\partial \boldsymbol{I}^{c_i}} \frac{\partial \boldsymbol{I}^{c_i}}{\partial \mathrm{L}^{c_i}} \left[ \begin{array}{cc} \frac{\partial \mathrm{L}^{c_i}}{\partial \delta \boldsymbol{x}^i} & \frac{\partial \mathrm{L}^{c_i}}{\partial \mathrm{L}^w} \frac{\partial \mathrm{L}^w}{\partial \delta \mathcal{O}} \end{array} \right] \tag{19}$$

The Jacobian matrix of the line reprojection error relative to $\boldsymbol{I}^{c_i}$ is expressed as:

$$\frac{\partial \boldsymbol{r}_l}{\partial \boldsymbol{I}^{c_1}} = \left[ \begin{array}{ccc} \frac{\partial r_1}{\partial l_1} & \frac{\partial r_1}{\partial l_2} & \frac{\partial r_1}{\partial l_3} \\ \frac{\partial r_2}{\partial l_1} & \frac{\partial r_2}{\partial l_2} & \frac{\partial r_2}{\partial l_3} \end{array} \right] =$$

$$\left[ \begin{array}{ccc} \frac{-l_1 \boldsymbol{s}_l^{\mathrm{T}} \boldsymbol{I}}{\left(l_1^2+l_2^2\right)^{\frac{3}{2}}} + \frac{u_s}{\left(l_1^2+l_2^2\right)^{\frac{1}{2}}} & \frac{-l_2 \boldsymbol{s}_l^{\mathrm{T}} \boldsymbol{I}}{\left(l_1^2+l_2^2\right)^{\frac{3}{2}}} + \frac{v_s}{\left(l_1^2+l_2^2\right)^{\frac{1}{2}}} & \frac{1}{\left(l_1^2+l_2^2\right)^{\frac{1}{2}}} \\ \frac{-l_1 \boldsymbol{e}_l^{\mathrm{T}} \boldsymbol{I}}{\left(l_1^2+l_2^2\right)^{\frac{3}{2}}} + \frac{u_e}{\left(l_1^2+l_2^2\right)^{\frac{1}{2}}} & \frac{-l_2 \boldsymbol{e}_l^{\mathrm{T}} \boldsymbol{I}}{\left(l_1^2+l_2^2\right)^{\frac{3}{2}}} + \frac{v_e}{\left(l_1^2+l_2^2\right)^{\frac{1}{2}}} & \frac{1}{\left(l_1^2+l_2^2\right)^{\frac{1}{2}}} \end{array} \right]_{2 \times 3} \tag{20}$$

According to the 3D line projection, Equation (21) is obtained.

$$\frac{\partial \boldsymbol{I}^{c_i}}{\partial \mathcal{L}^{c_i}} = \left[ \begin{array}{cc} \frac{\partial \boldsymbol{I}}{\partial \boldsymbol{n}} & \frac{\partial \boldsymbol{I}}{\partial \boldsymbol{d}} \end{array} \right] = \left[ \begin{array}{cc} \mathcal{K} & 0 \end{array} \right]_{3 \times 6} \tag{21}$$

The term $\frac{\partial \mathcal{L}^{c_i}}{\partial \delta \boldsymbol{x}^i}$ represents the Jacobian matrix of line features concerning translation and rotation errors in camera reference in which $\delta \boldsymbol{x}^i = \left[ \begin{array}{cc} \delta \boldsymbol{p} & \delta \boldsymbol{\theta} \end{array} \right]$. The line features only

optimize the translation $\boldsymbol{p}$ and the rotation $\boldsymbol{\theta}$ in the state variable, and the Jacobian matrix is calculated as follows:

$$
\begin{aligned}
\frac{\partial \mathcal{L}^{c_i}}{\partial \delta \boldsymbol{\theta}_{bb'}} &= \mathcal{T}_{bc}^{-1} \left[ \begin{array}{c} \partial\left(\boldsymbol{I} - [\delta\boldsymbol{\theta}_{bb'}]_\times\right)\boldsymbol{R}_{wb}^T\left(\boldsymbol{n}^w + [\boldsymbol{d}^w]_\times \boldsymbol{p}_{wb}\right)\partial\delta\boldsymbol{\theta}_{bb'} \\ \frac{\partial\left(\boldsymbol{I} - [\delta\boldsymbol{\theta}_{bb'}]_\times\right)\boldsymbol{R}_{wb}^T\boldsymbol{d}^w}{\partial\delta\boldsymbol{\theta}_{bb'}} \end{array} \right] \\
&= \mathcal{T}_{bc}^{-1} \left[ \begin{array}{c} \left[\boldsymbol{R}_{wb}^T\left(\boldsymbol{n}^w + [\boldsymbol{d}^w]_\times \boldsymbol{p}_{wb}\right)\right]_\times \\ \left[\boldsymbol{R}_{wb}^T\boldsymbol{d}^w\right]_\times \end{array} \right]_{6\times 3}
\end{aligned}
\tag{22}
$$

$$
\frac{\partial \mathcal{L}^{c_i}}{\partial \delta \boldsymbol{p}_{bb'}} = \mathcal{T}_{bc}^{-1} \left[ \begin{array}{c} \frac{\partial \boldsymbol{R}_{wb}^T\left(\boldsymbol{n}^w + [\boldsymbol{d}^w]_\times\left(\boldsymbol{p}_{wb} + \delta\boldsymbol{p}_{bb'}\right)\right)}{\partial\delta\boldsymbol{p}_{bb'}} \\ \frac{\partial \boldsymbol{R}_{wb}^T\boldsymbol{d}^w}{\partial\delta\boldsymbol{p}_{bb'}} \end{array} \right] = \mathcal{T}_{bc}^{-1} \left[ \begin{array}{c} \boldsymbol{R}_{wb}^T[\boldsymbol{d}^w]_\times \\ \boldsymbol{0} \end{array} \right]_{6\times 3}
\tag{23}
$$

The Jacobian matrix of the line in the camera reference concerning the line in the world reference is the inverse of the transformation matrix represented as Equation (24).

$$
\frac{\partial \mathcal{L}^{c_i}}{\partial \mathcal{L}^w} = \mathcal{T}_{wc}^{-1}
\tag{24}
$$

According to the orthogonal representation of a spatial line, the last term of Equation (19) can be defined as:

$$
\begin{aligned}
\frac{\partial \mathcal{L}^w}{\partial \delta \mathcal{O}} &= \left[ \begin{array}{cccc} \frac{\partial\mathcal{L}^w}{\partial\varphi_1} & \frac{\partial\mathcal{L}^w}{\partial\varphi_2} & \frac{\partial\mathcal{L}^v}{\partial\varphi_3} & \frac{\partial\mathcal{L}^w}{\partial\phi} \end{array} \right] \\
&= \left[ \begin{array}{cccc} \frac{\partial\mathcal{L}^v}{\partial\boldsymbol{U}}\frac{\partial\boldsymbol{U}}{\partial\varphi_1} & \frac{\partial\mathcal{L}^v}{\partial\boldsymbol{U}}\frac{\partial\boldsymbol{U}}{\partial\varphi_2} & \frac{\partial\mathcal{L}^w}{\partial\boldsymbol{U}}\frac{\partial\boldsymbol{U}}{\partial\varphi_3} & \frac{\partial\mathcal{L}^v}{\partial\boldsymbol{W}}\frac{\partial\boldsymbol{W}}{\partial\phi} \end{array} \right] \\
&= \left[ \begin{array}{cccc} 0 & -w_1 u_3 & w_1 u_2 & -w_2 u_1 \\ w_2 u_3 & 0 & -w_2 u_1 & w_1 u_2 \end{array} \right]_{6\times 4}
\end{aligned}
\tag{25}
$$

With the analysis of the Jacobians completed, iterative algorithms, such as Gaussian–Newton, can be employed to solve the local graph optimization of local mapping and global graph optimization of loop closure.

### 3.2. Loop Closure with Points and Lines

In human-made environments, the existence of weak textures (i.e., white wall) and frequent light changes leads to the false detection of the traditional BoW-based loop closure. The insufficient recognition of point features leads to the mismatch results between frames. To address this aspect, PL-BoW, a BoW that combines point and line features, is proposed, which utilizes the co-occurrence information and spatial proximity of visual words.

The visual words of point features are generated from the ORB descriptor, including the position $\boldsymbol{p}_{point}$ and direction $\theta_p$ of point features. At the same time, the direction $\theta_l$ and position information of the corresponding line feature is obtained from its endpoints. As shown in Figure 5, the combination of a point and a line is defined as a PL pair only when the direction and distance of the line and the point are close enough. To improve the search speed of PL pairs, a K-D tree is constructed by using the position of each point feature. Then the line features satisfying the PL pair are selected to build the point and line K-D tree.

Given a current keyframe $f_u$ and a candidate keyframe $f_c$, the corresponding BoW vectors are defined as $\left[v_{p1}, \ldots, v_{ps}, v_{l1}, \ldots, v_{lt}\right]$ and $\left[w_{p1}, \ldots, w_{ps'}, w_{l1}, \ldots, w_{lt'}\right]$, respectively. In loop closure detection, the similarity of two keyframes is calculated through the BoW vectors as follows:

$$
\begin{aligned}
S = &-\frac{N_p}{2(N_p + N_l)} \sum_{i,j}\left(\left|v_{pi} - w_{pj}\right| - \left|v_{pi}\right| - \left|w_{pj}\right|\right) \\
&-\frac{N_l}{2(N_p + N_l)} \sum_{i,j}\left(\left|v_{li} - w_{lj}\right| - \left|v_{li}\right| - \left|w_{lj}\right|\right)
\end{aligned}
\tag{26}
$$

where $N_p$ and $N_l$ represent the total number of point and line features in the image, respectively. The mismatched keyframes that have the lower score are removed from the set of candidate keyframes. The procedure of the improved PL-BoW based loop detection is given in Algorithm 1.

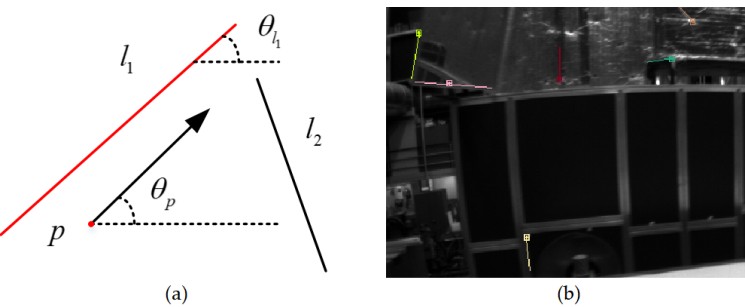

(a) (b)

**Figure 5.** Word pairs of points and lines. (**a**) Relative spatial co-occurrence of words in a pair. (**b**) Word pairs of point and line in a given image.

---

**Algorithm 1:** PL-BoW Based Loop Detection.

**Input**  : The keyframes set $F = \{f_1 \ldots f_i\}$, the KD-tree associated with $f_i$ and the current keyframe $f_u$;

**Output**: A revisit matching keyframe $f_{bm}$;

1    Select candidate keyframes through retrieving the words of points and lines in the image using Term Frequency-Inverse Document Frequency (TF-IDF);

2    $n_{Wi} = \text{NumberOfCommonView Words } (f_i)$;

3    $n_{PLi} = \text{NumberOfCommonViewPLpairs } (f_i)$;

4    **for** each $f_i \in F_c$ **do**

5        **if** $n_{Wi} < \text{Max}_{f_i \in F_c}\{n_{Wi}\}\&\&n_{PLi} < \text{Max}_{f_i \in F_c}\{n_{PLi}\}$ **then**

6        $f_i \cup F_{cm} \to F_{cm}$;

7        Calculate the similarity $S_i$;

8        $S_{\max} = \text{Max}\{S_i\}$;

9        **end**

10    **end**

11    **for** each $f_i \in F_c$ **do**

12        Remove $f_i$ with $S_i < 0.8S_{\max}$;

13    **end**

14    Perform space consistency detection on $F_{cm}$ to obtain $f_{bm}$.

---

## 4. Experimental Verification

To test and analyze the proposed PEL-SLAM, a set of experimental tests were performed with an Intel CPU i5-10060 KF@4.1GHz, 32GB RAM, without a dedicated GPU. OpenCV and g2o were mainly used as the environment running on an Ubuntu16.04 desktop. The proposed PL visual stereo SLAM algorithm was compared with popular methods, including ORB-SLAM2 [8], PL-SLAM [16], and stereo Point and Line based Visual Odometry (sPLVO) [23]. The above algorithms were tested on the KITTI stereo dataset [34] and EuRoC micro aerial vehicle (MAV) dataset [33], which provide several challenging sequences of images in indoor and outdoor environments. The absolute Root Mean Square Errors (RMSEs) between the estimated translation and rotation of the SLAM system and the groundtruth given in the dataset were computed by the EVO [35] evaluation tool.

### 4.1. Stereo SLAM on KITTI Dataset

The test results in terms of the RMES of ORB-SLAM2, PL-SLAM, and the proposed algorithm on the KITTI dataset are shown in Table 1. Since the strategy applied in the sPLVO is not appropriate for outdoor environments, sPLVO was not tested for comparison on KITTI dataset.

**Table 1.** Absolute RMSE errors of three methods on KITTI.

| Seg. | ORB-SLAM2 | | PL-SLAM | | PEL-SLAM | |
|---|---|---|---|---|---|---|
| | Trans. (m) | Rot. (deg) | Trans. (m) | Rot. (deg) | Trans. (m) | Rot. (deg) |
| 0 | 1.303313 | 0.018447 | 3.323087 | 0.068646 | **1.187783** | **0.017726** |
| 1 | **9.231962** | 0.025892 | 10.009194 | 0.033958 | 9.509255 | **0.024612** |
| 2 | 5.177579 | 0.027988 | 7.724952 | 0.065647 | **4.427225** | **0.026923** |
| 3 | **0.760564** | 0.014225 | 0.895579 | **0.011213** | 0.083256 | 0.013563 |
| 4 | 0.203649 | 0.457889 | 0.260442 | 0.431715 | **0.153068** | **0.322527** |
| 5 | **0.748097** | **0.008962** | 1.983561 | 0.022194 | 0.876509 | 0.009753 |
| 6 | 0.736519 | 0.011604 | 0.891829 | 0.031371 | **0.725683** | **0.010336** |
| 7 | 0.540521 | 0.010325 | 0.871499 | 0.038998 | **0.503406** | **0.010071** |
| 8 | 3.23028 | 0.028896 | 4.97701 | 0.030086 | **3.094443** | **0.025968** |
| 9 | 2.962183 | 0.029326 | 3.588782 | 0.031642 | **2.499952** | **0.027685** |

The lowest absolute translation and rotation errors for each test are marked in bold. Overall, the proposed method performs better than the other two methods. The translation and rotation errors of the proposed method are about 24.8% and 29.6% lower than that of ORB-SLAM2, respectively. However, it can be seen that the application of line features in the environment with insufficient line features hinders the accuracy of the SLAM process. As shown in Figure 6, the condition of the KITTI 05 dataset is mostly foresting with fewer line features, and the translation and rotation accuracy of the proposed method are reduced by 17.1% and 8.8%, respectively. Table 1 also shows that the proposed method outperforms PL-SLAM, which also employs both point and line features. One reason for this is that the use of PL-BoW suppresses the long-term positioning drift of the system in large outdoor scenes. Another reason for this result is that the proposed line feature outliers elimination method reduces the front-end mismatching and further improves the accuracy of the constraints constructed in the back-end optimization of the system.

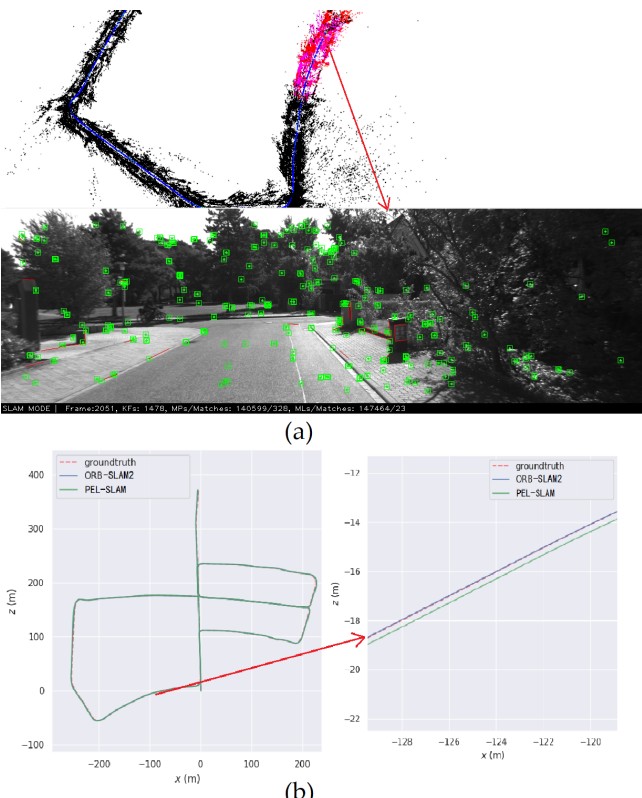

**Figure 6.** An example of the PEL-SLAM on KITTI 05. (**a**) Feature extraction and mapping of PEL-SLAM. (**b**) Trajectory comparison between ORB-SLAM2 and PEL-SLAM.

### 4.2. Stereo SLAM on EuRoC Dataset

The accuracy of the four algorithms (PEL-SLAM, PL-SLAM, sPLVO, and ORB-SLAM2) was also compared on the EuRoC dataset, which includes indoor environments images of machine halls and rooms. Table 2 shows the absolute translation and rotation errors of these four algorithms.

**Table 2.** Absolute RMSE errors of four methods on EuRoC.

| Seg. | PL-SLAM | | ORB-SLAM2 | | sPLVO | | PEL-SLAM | |
|---|---|---|---|---|---|---|---|---|
| | Trans. (m) | Rot. (deg) | Trans. (m) | Rot. (deg) | Trans. (m) | Rot. (deg) | Trans. (m) | Rot. (deg) |
| MH-01 | 0.156799 | 6.039926 | 0.038788 | 0.784221 | 0.039265 | 0.735803 | **0.037170** | **0.713027** |
| MH-02 | 0.142146 | 2.541990 | 0.051815 | 0.786260 | **0.042640** | 0.636756 | 0.047628 | **0.613111** |
| MH-03 | 0.146580 | 3.376991 | 0.039685 | 0.782628 | **0.037920** | **0.665110** | 0.043620 | 0.671286 |
| MH-04 | 0.123971 | 6.755803 | 0.131072 | 0.777158 | 0.063646 | 0.711528 | **0.059120** | **0.673853** |
| MH-05 | 0.554628 | 9.947981 | 0.091573 | 0.781631 | 0.054628 | 0.792583 | **0.047690** | 0.693294 |
| V1-01 | 0.168556 | 4.452180 | 0.087874 | 0.712945 | 0.087200 | 0.947981 | **0.082576** | **0.712721** |
| V1-02 | 0.168729 | 5.623589 | **0.064295** | 0.776845 | 0.067039 | 0.798271 | 0.064460 | **0.756860** |
| V1-03 | 0.419889 | 9.123150 | **0.069812** | **0.767960** | 0.070297 | 0.771652 | 0.085186 | 0.861888 |
| V2-01 | 0.194298 | 2.280268 | 0.085005 | 0.781473 | 0.072192 | 0.780698 | **0.063510** | **0.770080** |
| V2-02 | 0.251842 | 4.635829 | 0.056297 | 0.791928 | 0.065607 | 0.701221 | **0.054290** | **0.653020** |
| V2-03 | 0.567585 | 6.001996 | **0.272255** | 0.787116 | 0.372658 | 0.800124 | 0.405792 | **0.774410** |

It is clear from Table 2 that the strategy used by sPLVO for indoor line features makes it competitive in indoor environments. Nevertheless, the performance of the proposed method on the EuRoC dataset was generally superior to PL-SLAM, ORB-SLAM2, and sPLVO. Compared with the ORB-SLAM2, the translation and rotation accuracy of the proposed method are improved by nearly 5.7% and 7.3%, respectively. These improvements demonstrate that the incorporation of line features increases the accuracy of pose estimation and map construction. Although both sPLVO and the proposed method introduce line feature outliers elimination, the proposed method shows better accuracy in most testing cases. Compared with the state-of-the-art sPLVO, the maximum translation and rotation accuracy of the proposed method improves the results by 17.2% and 24.8%, respectively. These results indicate that using the entropy scale to measure the uncertainty of line features and applying PL-BoW to loop detection are beneficial to the accuracy of the SLAM process.

The tests also demonstrate that the proposed method environments with drastic changes and rapid motion of carriers. Figure 7 illustrates the estimated trajectory comparisons of MH-02 and V1-01, which contains the results with typical rapid motion and changing light. The accuracy of the two methods based on point and line features (i.e., sPLVO and PEL-SLAM) is higher than the methods that use only-point-features, ORB-SLAM2. Meanwhile, the average rotation accuracy of the proposed method is nearly 14.3% higher than that of sPLVO.

### 4.3. Comparison of Processing Time

Due to the fact that the real-time performance is one of the important indicators of the SLAM process, a comparison between the average processing time per frame of different methods was performed (Table 3). Table 3 shows that the operating time of the SLAM process is directly affected by the image resolution, which means that the process needs more time to process the KITTI images. At the same time, the application of line features increases the running time of the system, especially the detection of line features. The PEL-SLAM reduces the time consumption during the line features detection. Compared with PL-SLAM, the proposed method saves approximately 7.7% and 12.2% of the average running time in KITTI and EuRoC datasets, respectively. The running time of the proposed method is 4.5% faster than that of sPLVO without sacrificing the accuracy of the pose estimation and map construction.

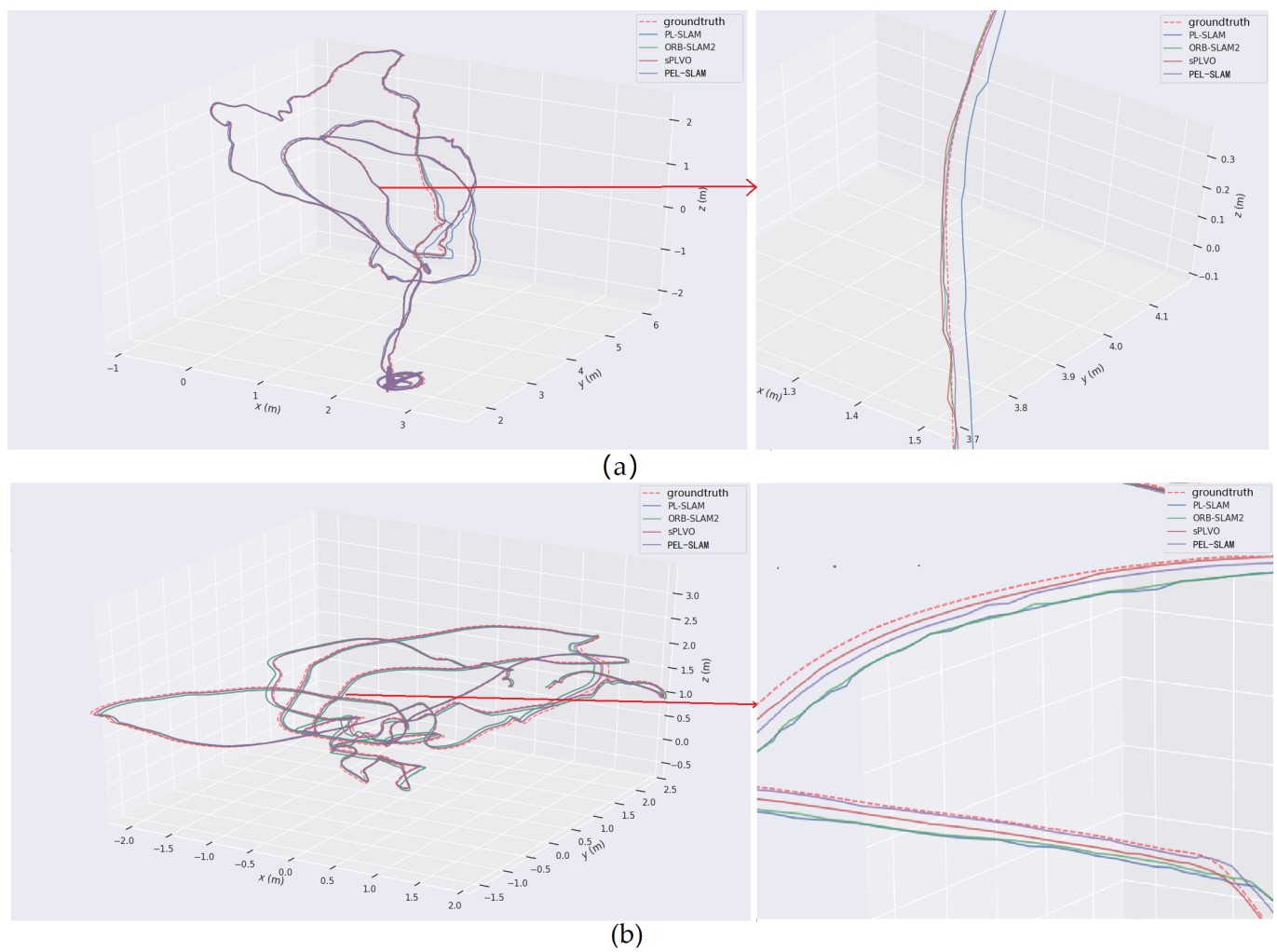

**Figure 7.** Trajectory comparison between PL-SLAM, ORB-SLAM2, sPLVO, and the proposed PEL-SLAM. (**a**) The 3D view of the estimated trajectories on EuRoC MH-02. (**b**) The 3D view of the estimated trajectories on EuRoC V1-01.

**Table 3.** Average processing time per frame on KITTI and EuRoC.

| KITTI Seg. | ORB-SLAM Time (s) | PL-SLAM Time (s) | PEL-SLAM Time (s) | EuRoC Seg. | ORB-SLAM Time (s) | PL-SLAM Time (s) | sPLVO Time (s) | PEL-SLAM Time (s) |
|---|---|---|---|---|---|---|---|---|
| 0 | 0.05932 | 0.09126 | 0.07917 | MH-01 | 0.04481 | 0.08069 | 0.08553 | 0.07410 |
| 1 | 0.08233 | 0.09912 | 0.09119 | MH-02 | 0.04333 | 0.08232 | 0.08238 | 0.07798 |
| 2 | 0.06338 | 0.08637 | 0.07817 | MH-03 | 0.04346 | 0.08327 | 0.07756 | 0.07635 |
| 3 | 0.06470 | 0.08653 | 0.07846 | MH-04 | 0.03610 | 0.07563 | 0.07299 | 0.06956 |
| 4 | 0.06627 | 0.09610 | 0.08687 | MH-05 | 0.03747 | 0.07267 | 0.06968 | 0.06660 |
| 5 | 0.06738 | 0.10634 | 0.08765 | V1-01 | 0.03360 | 0.07085 | 0.06423 | 0.06133 |
| 6 | 0.07303 | 0.09923 | 0.09506 | V1-02 | 0.03548 | 0.08065 | 0.05815 | 0.06585 |
| 7 | 0.06103 | 0.08712 | 0.08899 | V1-03 | 0.03299 | 0.07695 | 0.06187 | 0.05910 |
| 8 | 0.06534 | 0.09138 | 0.08847 | V2-01 | 0.03232 | 0.06924 | 0.06611 | 0.06172 |
| 9 | 0.06156 | 0.09175 | 0.08749 | V2-02 | 0.03731 | 0.07138 | 0.06765 | 0.05856 |

## 5. Conclusions

In this paper, a method that can extract point and line features with the purpose of using them to improve the positioning, mapping, and loop detection is proposed. The PEL-SLAM system solves the problem of point-based failure in poorly textured scenes. The proposed method uses the faster EDlines instead of the widely used LSD and leverages the entropy scale to measure the uncertainty of line features. Meanwhile, the proposed PL-BoW is constructed and applied to the loop detection, which improves the accuracy

of loop keyframe matching. Such mechanisms system enables SLAM to be performed in real-time without loss of accuracy while producing a complete point-linemap. Finally, the detailed experiments on KITTI and EuRoC datasets show that the proposed method outperforms many well-known methods with respect to translation and rotation accuracy in challenging environments, with the additional benefit of using less time consumption compared with the state-of-the-art PL SLAM systems. In the proposed algorithm, the accuracy of the system depends on the performance of the detected visual features. The way of combining other sensors information to improve the robustness of the point-line SLAM will be the focus of future research.

**Author Contributions:** H.R., A.R.-S. and Y.G. conceived the idea; H.R., X.X. and Y.Z. designed the software, collected the data, and analyzed the experimental data; L.G. collected the related resources and supervised the experiment; A.R.-S. proposed the comment for the paper and experiment. All authors have read and agreed to the published version of the manuscript.

**Funding:** This work is sponsored by the National Natural Science Foundation of China (NSFC. 61803118), the Special Project of Chongqing Technology Innovation and Application Development (cstc2019jscx-msxmX0423), the Science and Technology Research Program of Chongqing Municipal Education Commission (KJZD-K201804701), and the Post Doc. Foundation of Heilongjiang Province (LBH-Z17053).

**Data Availability Statement:** The data that support the findings of this study are available from the corresponding author upon request.

**Conflicts of Interest:** The authors declare no conflict of interest.

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
