# Peer review of "Point-Line Visual Stereo SLAM Using EDlines and PL-BoW"

_remotesensing, doi:10.3390/rs13183591_

Round 1
Reviewer 1 Report
This work presents a sound approach to stereo SLAM with the fusion of several techniques such as line features, also combined with point features, and the basic hierarchy of word diccionary of BOW already exploited by ORB-SLAM models, which is in fact the main structure used in this work. All in all, despite the fact they are all well-acknowledged frameworks and techniques, the combination, specially enhanced by the use of line features, makes the reader set aside the lack of novelty to concentrate more on the contrinution itself of the final application, soundness of the research structure and implementation. Results are valid, benchmarked against publicly available datasets (outdoors which is also fine) and compared again the basic ORB-SLAM framework.
Nonetheless, I would like to point out the following comments for authors to address in their next version:
Authors should discuss more on wider field of view cameras with high distortion. Despite this fact, they provide important benefits to cover wider scenes by simply using a mono camera. Why then using planar, and stereo system? Please address these two aspects.
In the same sense, I miss more related work related to FOV cameras, like fish eye, panoramic, ominidirectional and so on. Plus others on feature lines. Besides, other sort of descriptors (global appearance/holistic) can provide applications to estimate, for instance, height. Please investigate more on topics like the following, to complete the related work and justify the improvement of your approach in contrast to the following, but specially in the selection of the system, that is: planar stereo system.
https://scholar.google.com/citations?view_op=view_citation&hl=en&user=SC9wV2kAAAAJ&cstart=100&pagesize=100&sortby=pubdate&citation_for_view=SC9wV2kAAAAJ:YsMSGLbcyi4C
https://ieeexplore.ieee.org/document/4650741
https://www.mdpi.com/2072-4292/11/3/323
"Benefits of large FOV cameras for visual odometry". http://rpg.ifi.uzh.ch/fov.html
Minor:
pag 2. line 86. EDlines has not yet been defined specifically. Reference?
Reviewer 2 Report
Dear Authors,
Please find the attached file for my comments. Please update the paper based on my comments and resubmit it.
Best Regards

Round 2
Reviewer 2 Report
Dear Authors,
Thank you for addressing all my comments and the paper is accepted from my side for publication.
Best Regards